# CASA: Category-agnostic Skeletal Animal Reconstruction

**Yuefan Wu**[1]* **Zeyuan Chen**[1]* **Shaowei Liu**[2] **Zhongzheng Ren**[2] **Shenlong Wang**[2]
[1] University of Science and Technology of China [2] University of Illinois Urbana-Champaign
https://Iven-Wu.github.io/CASA

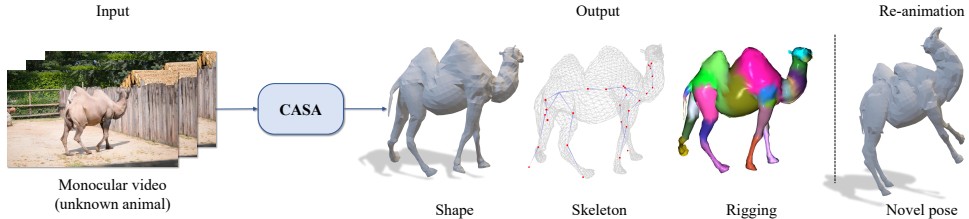

Figure 1: Given a monocular video with an animal from unknown category, CASA jointly infers the articulated skeletal shape and rigging through optimization, which can be animated into novel poses.

## Abstract

Recovering the skeletal shape of an animal from a monocular video is a longstanding challenge. Prevailing animal reconstruction methods often adopt a control-point driven animation model and optimize bone transforms individually without considering skeletal topology, yielding unsatisfactory shape and articulation. In contrast, humans can easily infer the articulation structure of an unknown animal by associating it with a seen articulated character in their memory. Inspired by this fact, we present **CASA**, a novel **C**ategory-**A**gnostic **S**keletal **A**nimal reconstruction method consisting of two major components: a video-to-shape retrieval process and a neural inverse graphics framework. During inference, CASA first retrieves an articulated shape from a 3D character assets bank so that the input video scores highly with the rendered image, according to a pretrained language-vision model. CASA then integrates the retrieved character into an inverse graphics framework and jointly infers the shape deformation, skeleton structure, and skinning weights through optimization. Experiments validate the efficacy of CASA regarding shape reconstruction and articulation. We further demonstrate that the resulting skeletal-animated characters can be used for re-animation.

## 1 Introduction

Recovering the shape, articulation, and dynamics of animals from images and videos is a longstanding task in computer vision and graphics. Achieving this goal will enable numerous future applications for 3D modeling and reanimation of animals. Nevertheless, accurately reasoning about the geometry and kinematics of animals in the wild remains an ambitious problem for three reasons: 1) *partial visibility* of the captured animal, 2) *variability* of shape across different categories, and 3) *ambiguity* of unknown kinematics. Take the camel in Fig. 1 as an example – to faithfully model it requires: 1) hallucinating the unobserved (*e.g.*, occluded, back-side) region, 2) recovering its unique shape and scale, and 3) predicting its kinematic structure.

Remarkable progress has been made in addressing the aforementioned challenges by either exploiting richer sensor configurations (*e.g.*, multi-view cameras [11, 17] or depth sensors [41, 42, 50]), or by

---

*Indicates equal contribution

36th Conference on Neural Information Processing Systems (NeurIPS 2022).

making strong class-specific assumptions (*e.g.*, humans [34, 55–57, 78, 80] or quadruped animals [83–85]). However, these assumptions greatly limit the applicability of reconstruction systems, which fail to generalize to animals in the wild.

We aim to reconstruct *arbitrary articulated animals using monocular videos casually captured in the wild*. Recent works [74–76, 70, 45] demonstrate promising results. However, they often impose non-realistic assumptions on articulation, such as control point driven deformation [74–76, 70] or free-form deformation [45, 50]. As a result, they fall short of the goal of modeling skeletal characters that can be realistically re-animated in downstream applications. Furthermore, there remains significant improvement space for the quality of the inferred animal shape.

In this work, we propose **CASA**, a novel solution for **C**ategory-**A**gnostic **S**keletal **A**nimal reconstruction in the wild. CASA jointly estimates an arbitrary animal's 3D shape, kinematic structure, rigging weight, and articulated poses of each frame from a monocular video (Fig. 1). Unlike existing nonrigid reconstruction works [74, 75, 70], we exploit a skeleton-based articulation model as well as forward kinematics, ensuring the realism of the resulting skeletal shape (§ 3.1). Specifically, we propose two novel components: a video-to-shape retrieval process and a skeleton-based neural inverse graphics framework. Given an input video, CASA first finds a template shape from a 3D character assets bank so that the input video scores highly with the rendered image, according to a pretrained language-vision model [47] (§ 3.2). Using the retrieved character as initialization, we jointly optimize bone length, joint angle, 3D shape, and blend skinning weight so that the final outputs are consistent with visual evidence, *i.e.*, input video (§ 3.3).

Another issue that hinders the study of animal reconstruction is that the existing datasets [4, 13, 31] lack realistic video footage and ground-truth labels across different dynamic animals. To address it, we introduce a photo-realistic synthetic dataset **PlanetZoo**, which is generated using the physical-based rendering [7] and rich simulated 3D assets [35]. PlanetZoo is large-scale and consists of 251 different articulated animals. Importantly, PlanetZoo provides ground-truth 3D shapes, skeletons, joint angles, and rigging, allowing evaluation of category-agnostic 4D reconstruction holistically (§ 4).

We evaluate CASA on both PlanetZoo and the real-world dataset DAVIS [46]. Experiments demonstrate that CASA recovers fine shape and realistic skeleton topology, handles a wide variety of animals, and adapts well to unseen categories. Additionally, we showed that CASA reconstructs a skeletal-animatable character readily compatible with downstream re-animation and simulation tasks.

In summary, we make the following contributions: 1) We propose a simple, effective, and generalizable video-to-shape retrieval algorithm based on a pretrained CLIP model. 2) We introduce a novel neural inverse graphics optimization framework that incorporates stretchable skeletal models for category-agnostic articulated shape reconstruction. 3) We present a large-scale yet diverse skeletal shape dataset PlanetZoo.

## 2  Related work

**Category-specific nonrigid reconstruction.** Parametric model (template) is a classic approach for category-specific nonrigid reconstruction[40, 18, 2, 78, 77, 36, 32, 54] . Templates are built for various objects including human body [34, 1], hand [53], face [29, 3], and animals [85, 2]. Template-free methods [19, 30, 25, 64, 57, 70, 23, 44, 69] study the problem of predicting nonrigid objects directly for a specific category. However, these methods heavily rely on strong category priors such as key-points annotations [19], canonical shapes [10], temporal consistency constraints [30], or canonical surface mapping [23], making them hard to generalize to category-agnostic setting.

**Category-agnostic nonrigid reconstruction.** Classic approaches [5, 24, 60, 43] utilize Non-Rigid Structure from Motion (NRSfM) for non-rigid reconstruction, which requires point correspondences and strong priors on shape or motion. However, NRSfM does not work well for monocular videos in the wild, in which obtaining long-range correspondences is hard. Recent works adopt learning-based approaches via either dynamic reconstruction [27, 50] or inverse graphics optimization [74–76].

However, these methods do not reason skeletal kinematics and suffer from incorrect shape prediction when certain parts are occluded or invisible.

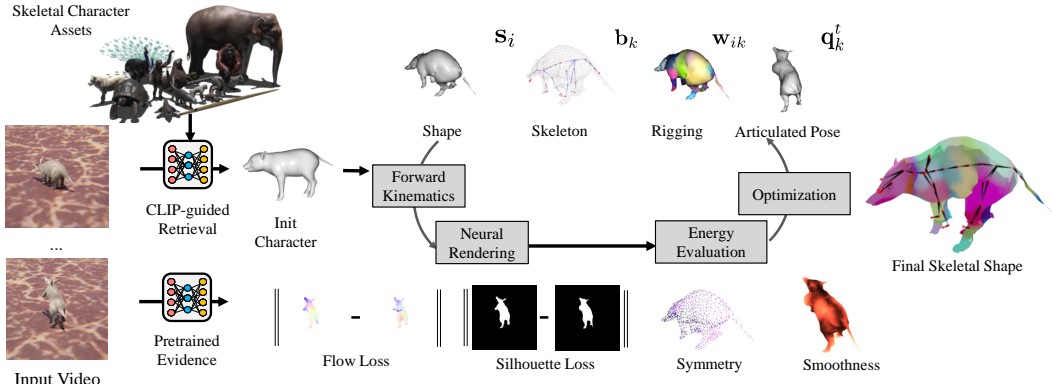

Figure 2: **Overview.** Given an input video, A video-to-shape retrieval process is first conducted with the guidance of pre-trained CLIP (§ 3.2). Initialized by the retrieved shape, we jointly optimize shape, skeleton (bone length and joint angle), and skinning through inverse rendering (§ 3.3).

**Animatable shape.** Humans and animals undergo skeleton-driven deformation. A skeletal shape often consists of two parts: a surface mesh representation used to draw the character and a skeleton structure to control the motion. Each skeleton is a hierarchical set of bones. Each bone has associated with a portion of vertices (skinning). The bone's transformation is determined through a forward kinematics process. As the character is animated, the bones change their transformation over time. Linear Blend Skinning (LBS) [37, 28] is a standard way for modeling skeletal deformation, which deforms each vertex of the shape based on a linear combination of bone transformations. For improvement, multi-weight enveloping [68, 38] is used to overcome the issue of shape collapse near joints when related bones are rotated or moved. Dual-quaternion blending (DQB) [20] adopts quaternion rotation representation to solve the artifacts in blending bone rotations, and STBS [15] extends LBS to include the stretching and twisting of bones. Recent works [74–76] employs LBS for modeling the motion of shapes recovered from video clips. However, these methods do not enforce a skeleton-based forward kinematic structure. Hence their recovered animated shapes are not interpretable and cannot be directly used in skeletal animation and simulation pipeline.

**Language-vision approaches.** Self-supervised language-vision models have gone through rapid advances in recent years [62, 66, 48] due to their impressive generalizability. The seminal work CLIP [66] learns a joint language-vision embedding using more than 400 billion text-image pairs. The learned representation is semantically meaningful and expressive, thus has been adapted to various downstream tasks [82, 71, 49, 79, 65]. In this work, we adopt CLIP in the retrieval process.

**3D reconstruction dataset.** A plethora of synthetic 3D datasets [61, 81, 16, 52] and interactive simulated 3D environments [58, 59, 9, 67] have been proposed in recent years. ShapeNet [8] provides a benchmark for common static 3D object shape reconstruction. Among all of these datasets, only a few aim for dynamic object reconstruction [31]. Given that the synthetic dataset has a large domain gap to real-world settings, photo-realistic rendered samples are strongly preferred. To this end, we propose a photo-realistic synthetic dataset PlanetZoo to study the dynamic animal reconstruction problem. PlanetZoo contains high-fidelity assets and covers a wide range of animal categories.

## 3 Category-agnostic skeletal animal reconstruction

CASA aims to reconstruct various dynamic animals in-the-wild from monocular videos. It takes as input the RGB frames $\{\mathbf{I}_t\}_{1...T}$ from a monocular video, object masks $\{\mathbf{M}_t\}_{1...T}$, and optical flow maps $\{\mathbf{F}_t\}_{1...T}$ computed from consecutive frames. From these inputs, we aim to recover an animal shape $\mathbf{s}_0$ and its articulated deformed shape $\mathbf{s}_t$ at each time $t$.

Fig. 2 demonstrates an overview of our approach. Our method exploits the skeletal articulation model and forward kinematics to ensure the realism of the resulting skeletal shape (§ 3.1). Our method consists of two phases. In the retrieval phase (§ 3.2), CASA finds a template character from an existing asset bank based on the similarity to the input video in the deep embedding space using a pre-trained encoder. The retrieved template is fed into the neural inverse graphics phase as

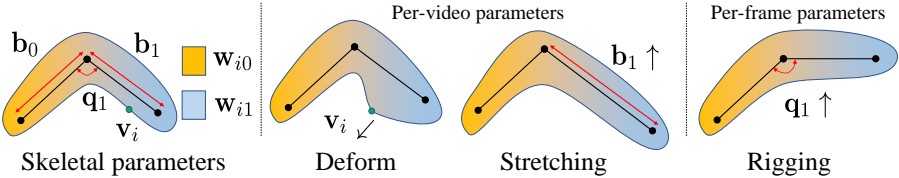

Figure 3: **Skeletal shape parametric model.** Our parametric model consists of joint angles $\mathbf{q}$, bone length $\mathbf{b}$, skinning weight $\mathbf{w}$, and vertex positions $\mathbf{v}$. Joint angles $\mathbf{q}$ change per-frame and the others are universal. Utilizing this model, animal shapes are tuned by vertex deformation as well as the stretching of bones. The target articulation would be fit by predicting per-frame joint angles.

initialization(§ 3.3). Finally, we jointly reason the final shape, skeleton, rigging and the articulated pose of each frame through an energy minimization framework.

## 3.1 Articulation model

**Skeletal parameters.** We exploit a stretchable bone-based skeletal model for our deformable shape parameterization ,as shown in Fig. 3. This articulated model consists of three components: 1) a triangular mesh consisting of a set of vertices and shapes in the canonical pose, describing the object's shape; 2) a set of bones connected by a kinematic tree. Each one has a bone length parameter and associated rigging weight over each vertex; 3) a joint angle describing the relative transformation between each adjacent bone. To summarize, a deformed shape $\mathbf{s}_t$ at time $t$ can be represented as:

$$\mathbf{s}_t = \left\{ \left\{ \mathbf{v}_i \in \mathbb{R}^3 \right\}_N, \left\{ \mathbf{f}_j \in \mathbb{I}^3 \right\}_T, \left\{ \mathbf{w}_i \in \Delta^K \right\}_N, \left\{ b_k \in \mathbb{R}^1 \right\}_K, \left\{ \mathbf{q}_k^t \in \mathbb{SO}(3) \right\}_K \right\} \quad (1)$$

where $\mathbf{v}_i$ is the vertex position, $\mathbf{f}_j$ represents a triangle face parameterized as a tuple of three vertex indices, $\mathbf{w}_i$ is the rigging weight constrained in a K-dimensional simplex: $\Delta^K = \left\{ (w_0, \ldots, w_K) \in \mathbb{R}^K \mid \sum_{k=0}^K w_k = 1 \text{ and } w_k \geq 0 \right\}$; $b_k$ is the scalar bone length for each bone and $\mathbf{q}_k^t$ is the joint angle for each bone at a time $t$, represented as a unit quaternion in the $\mathbb{SO}(3)$ space. Note that all the variables except joint angles are shared across time.

Our skeletal model is grounded by the nature of many articulated objects. Unlike the commonly used control-point based articulation [41, 75, 74], our model reflects the constraints imposed by bones and joints, leading to more natural articulated deformation. Compared to category-specific parametric models [34, 85], it is more flexible and generalizable.

**Forward kinematics.** We use the forward kinematic model [26] to compute the transformation of each bone along the kinematic tree. Specifically, the rigid transformation of a bone is uniquely defined by the joint angles and bone length scales of the bone itself and its ancestors along the kinematic tree. Given a bone $k$, the rigid transform between its own frame and the root can be computed by recursively applying the relative rigid transformation along the chain:

$$\mathbf{T}_k^t(\mathbf{q}_{\text{ans}(k)}^t, \mathbf{b}_{\text{ans}(k)}) = \mathbf{T}_{\text{pa}(k)}^t \mathbf{T}_{k,\text{pa}(k)}^t(\mathbf{q}_k^t, b_k) = \prod_{k' \in \text{ans}(k)} \mathbf{T}_{k',\text{pa}(k')}^t(\mathbf{q}_{k'}^t, b_{k'}) \quad (2)$$

where $\mathbf{T}_k^t, \mathbf{T}_{\text{pa}(k)}^t$ is the transformation of bone $k$ and its parent node at frame $t$. $\mathbf{q}_{\text{ans}(k)}, \mathbf{b}_{\text{ans}(k)}$ are the joint angles and bone lengths from all the ancestors of target bone $k$. $\mathbf{T}_{k,\text{pa}(k)}^t(\mathbf{q}_k^t, b_k)$ is the relative rigid transform between the bone and its parent frame, consists of a joint center translation along z axis decided by the bone length $b_k \mathbf{e}_z = [0, 0, b_k]$ and a rotation around the joint center $\mathbf{R}(\mathbf{q}_k^t)$.

**Linear Blend Skinning (LBS).** Given the rigid transformation of each bone, we adopt LBS to compute the transform of each point. Specifically, the deformation of each point $\mathbf{v}$ is decided by a linear composing the rigid transforms of bones through its skinning weight: $\mathbf{v}_i^t = \sum_{k=0}^K w_{i,k} \mathbf{T}_k^t \mathbf{v}_i$ where $w_{i,k}$ is the rigging weight. $\mathbf{T}_k^t$ is the transformation of bone $k$ at frame $t$ as defined in Eq. 2. For simplicity, we omit its dependent variables.

**Stretchable bones.** The aforementioned nonrigid deformation assumes independent shape and bone lengths. Naturally, the shape should deform accordingly as bone length changes. To model such a relationship, we exploit a stretchable bone deformation model [15]. For each bone, we compute an

additional scaling transform along the bone vector direction based on its bone length change and apply this scaling transform to any points associated with this bone. Fig. 3 provides an illustration of stretching. For mathematical details on stretching, we refer readers to our supplementary document.

Stretchable bones bring a two-fold advantage: 1) it allows us to model shape variations within the same topological structures; 2) it makes the topological structure adjustable by "shrinking" the bones, e.g., a quadrupedal animal can be evolved into a seal-like skeletal model through optimization.

**Reparameterization.** Optimizing each vertex position offers flexibility yet might lead to undesirable mesh due to the lack of regularization. We use a neural displacement field to re-parameterize the vertex deformation to incorporate regularization, such as smooth and structured vertex deformation. We use a coordinate-based multi-layer perceptron (MLP) to define this displacement field $\mathcal{V}$. In addition, we incorporate a global scale scalar $\mathbf{u}$ in our framework to handle the shape misalignment between our initialization and the target. The position of each vertex is defined as:

$$\mathbf{v}'_i = u\mathbf{v}_i + \mathcal{V}_\theta(u\mathbf{v}_i) \tag{3}$$

where $u\mathbf{v}_i$ is the scaled position at the canonical pose, $\mathcal{V}_\theta(u\mathbf{p}_i)$ represents the vertex offset parameterized by $\theta$, and $\mathbf{v}'_i$ is the updated position for vertex $i$. During inference, instead of directly optimizing $\mathbf{v}_i$, we will only optimize the parameters of the displacement network. This displacement field reparameterization allow us to smoothly deform canonical shape during inference. In practice we find this implicit regularization compares favorably over explicit smoothness terms such as as-rigid-as-possible and laplacian smoothness.

## 3.2 Video-to-shape retrieval

Directly optimizing all skeletal parameters defined in Eq. 1 is challenging due to the variability and highly structured skeletal parameterization. Many deformable objects share similar topology structures, albeit with significant shape differences. To address it, we propose initializing skeletal shapes in a data-driven manner by choosing the best matching template from an asset bank.

Videos and skeletal shapes are in two different modalities; hence establishing a similarity measure is hard. Inspired by the recent success of language-vision pretraining models for 3D [66, 39], we utilize realistic rendering and pretraining image embedding models [47] to bridge this gap. Specifically, we first pre-render video footage for each character in the asset bank through a physically based renderer [7]. The environment lighting, background, articulated poses, and camera pose for each video are randomized to gain diversity and robustness. We then extract the image embedding features for each video using CLIP [47], which is a language-vision embedding model that is pretrained on a large-scale image-caption dataset. It captures the underlying semantic similarity between images well despite the large appearance and viewpoint difference. This is particularly suitable for retrieving shapes with similar kinematic structures, as the kinematics of animals is often related to semantics.

During inference, we extract embedding features from a given input video and measure the L2 distance between the input video and the rendered video of each object in the embedding space. We then select the highest-scoring articulated shape as the retrieved character. Please refer to the supplementary material for implementation details.

## 3.3 Neural inverse graphics via energy minimization

We expect our final output skeletal shape to 1) be consistent with the input video observation; 2) reflect prior knowledge about articulated shapes ,such as symmetry and smoothness. Inspired by this, we formulate ours as an energy minimization problem.

**Energy formulation.** We exploit visual cues from each frame including segmentation mask $\{\mathbf{M}_t\}$ and optical flow $\{\mathbf{F}_t\}$ through off-the-shelf neural networks [63, 74]. We also incorporate two types of prior knowledge, including the motion smoothness for joint angles $\mathbf{q}_k^t$ of bones across frames, as well as the symmetry constraint for the per-vertex offset $\mathbf{v}_i$ at the reference frame. In particular, let $\mathcal{I} = (\{\mathbf{I}_t\}, \{\mathbf{M}_t\}, \{\mathbf{F}_t\})$ be the input video frames and corresponding visual cues, and $\{\mathbf{s}_t\}$ be the predicted shapes of all frames, we formulate the energy for minimization as:

$$\min_{\{\mathbf{s}_t\}_{1\dots T}} \lambda_1 E_{\text{cue}}(\{\mathbf{s}_t\}, \mathcal{I}) + \lambda_2 E_{\text{smooth}}(\{\mathbf{s}_t\}) + \lambda_3 E_{\text{symm}}(\{\mathbf{s}_t\}) \tag{4}$$

where $E_{\text{cue}}$ measures the consistency between the articulated shape and the visual cues; $E_{\text{smooth}}$ promotes smooth transitions overtime; $E_{\text{symm}}$ encodes the fact that many articulated objects are

symmetric. The three energy terms complement each other, helping our model capture motions of the object according to visual observations, as well as constraining the deformed object to be natural and temporally consistent. We describe details of each term below.

**Visual cue consistency.** The visual cue consistency energy measures the agreement between the rendered maps of the articulated shape and 2D evidence (flow and silhouettes) from videos. We use a differentiable renderer [33] to generate projected object silhouettes $\{\mathbf{M}_t\}$. Additionally, we project vertices of predicted shapes at two consecutive frames to the camera view and compute the projected 2D displacement to render optical flow, following the prior work [74]. We leverage PointRend [22] for object segmentation and volumetric correspondence net [73] for flow prediction. The energy measures the difference between the rendered and inferred cues in $\ell_2$ distance:

$$E_{\text{cue}} = \sum_t \left\{ \|\mathbf{M}_t - \pi_{\text{seg}}(\mathbf{s}_t)\|^2 + \beta \|\mathbf{F}_t - \pi_{\text{flow}}(\mathbf{s}_t)\|^2 \right\} \tag{5}$$

where $\beta$ is the trade-off weight; $\pi_{\text{flow}}(\mathbf{s}_t)$ is the rendered the 2D flow map for each pixel given the deformed shape $\mathbf{s}_t$; $\pi_{\text{seg}}(\mathbf{s}_t)$ is the rendered object mask. Similar to previous inverse rendering work [75], the object-camera transform is encoded as the root node transform.

**Motion smoothness.** This energy term encodes that motions of animals should be smooth and continuous across frames. We impose a constraint to ensure that there is little difference in joint angles of one bone from two consecutive frames should be slight. We implement this by computing the multiplication of the joint quaternion at the current frame and the inverse of the joint quaternion at the next frame, which should be close to an identity quaternion $\mathbf{q} = (0, 0, 0, 1)$:

$$E_{\text{smooth}} = \sum_t \sum_k \|(\mathbf{q}_k^t)^{-1} \circ \mathbf{q}_k^{t+1} - \mathbf{q}\|^2 \tag{6}$$

where $\circ$ is the quaternion composition operator.

**Symmetry offset.** This term encourages the resulting shape in canonical space to be symmetric at the reference frame. It is inspired by the fact that most animals in the real world are symmetric if putting them into a certain canonical pose (e.g., 'T-pose' for bipedal animals). Following previous works [74], we enforce this property at the canonical shape (where the joint angles are all zero). Specifically, we calculate the chamfer distance for measuring the similarity between the set of vertices $\{\mathbf{v}_i\}$ under the canonical shape and its reflection:

$$E_{\text{symm}} = \mathcal{L}_{\text{cham}}((\{\mathbf{v}_i\}), \mathbf{H}(\{\mathbf{v}_i\})) \tag{7}$$

where $\mathbf{H}$ is the Householder reflection matrix.

### 3.4 Inference

We reason the skeletal shape by minimizing the energy defined in Eq. 4. Our optimization variables include the vertex position $\mathbf{v}_i$ at the canonical pose, the rigging weight $\mathbf{w}_i$, bone length $\mathbf{b}_k$ as well as the joint angles $\mathbf{q}_k^t$. All the variables except joint angles are shared across time, and joint angles are optimized per frame.

**Initialization.** We initialize the vertex position, bone length, and the rigging weight $\mathbf{v}_i, \mathbf{b}_k, \mathbf{w}_i$ of canonical shape using the retrieved template character. The skeleton tree structure of this character is also taken as the basis of our skeletal parameterization. We initialize all the joint rotations as a unit quaternion.

**Optimization.** The energy function is fully differentiable and can be optimized end-to-end. We use Adam optimizer [21] to learn all the optimization variables. We adopt a scheduling strategy to avoid getting stuck at a local minimum. Specifically, we first optimize the mesh scaling factor based on silhouette correspondences. We then jointly update the bone length scale, joint angles, and the neural offset field by minimizing the energy function defined in Eq. 4.

## 4 PlanetZoo dataset

Benchmarking 4D animal reconstruction requires a large amount of nonrigid action sequences with ground truth articulated 3D shapes. Towards this goal, we construct a synthetic dataset PlanetZoo

Table 1: Quantitative results on PlanetZoo.

| Method | mIoU ↑ | mCham ↓ | Skinning ↓ | Joint ↓ | Re-animation ↓ |
|--------|--------|---------|------------|---------|----------------|
| ViSER [75] | 0.190 | 0.133 | 6.398 | 0.416 | 0.028 |
| LASR [74] | 0.306 | 0.094 | **2.576** | 0.158 | 0.033 |
| CASA (ours) | **0.512** | **0.053** | 3.288 | **0.089** | **0.010** |

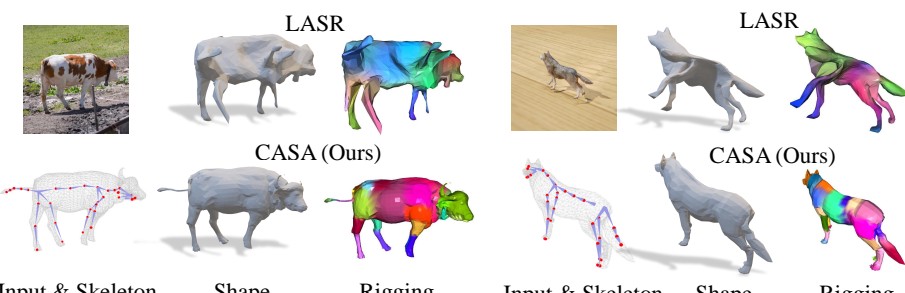

Figure 4: **Reconstructed articulated shape.** Top-left: input; top: LASR; bottom: ours.

consisting of hundreds of animated animals with textures and skeletons from different categories. Appendix Fig. **??** depicts a snapshot of the assets and rendering images from our dataset, demonstrating the diversity and quality.

**Data generation.** We extracted animal meshes from the zoo simulator Planetzoo [35]. Cobratools [12] are used to extract those meshes along with their skeleton. With the extracted mesh and skeleton, we further render RGB maps, segmentation masks and optical flow for each frame using Blender.

**Assets.** To set a diverse environment, we set the background with random HDRI pictures for environmental lighting, and set floor textures with random materials from ambientCG[2]. To reduce the gap between synthetic and real, we set the location of the light source along with its strength to simulate different environments in real-world situations.

**Camera.** To generate realistic action sequences, we randomly change camera locations between every 12 frames, resulting in a constant view-point change following the animal. The camera is allowed to rotate at a certain angle, ranging from $15°$ to $22.5°$.

**Articulation.** In order to obtain animated animal sequences, for every 12 frames, 8 bones are selected from the skeleton tree with a transformation attached. The angle value of rotation for each bone is sampled from a uniform distribution. By doing this, we are able to cover the whole action space, providing diverse action sequences.

**Rendering.** We generate silhouettes, optical flow, depth map, camera parameters, and RGB images using Vision Blender [7]. The physically based renderer is capable of showing fine details such as fur, making the rendering results more realistic. For each animal in the dataset, 180 frames are rendered.

## 5 Experiments

In this section, we first introduce our experimental setting (§ 5.1). We then compare CASA against a comprehensive set of articulated baselines in various reconstruction and articulation metrics on both simulation (§ 5.2) and real-world datasets (§ 5.3). Finally, we demonstrate our inferred shape can be used for downstream reanimation tasks (§ 5.4).

### 5.1 Experimental setup

**Benchmarks.** We validate our proposed method on two datasets. Our proposed photorealistic rendering dataset PlanetZoo as well as a real-world animal video dataset DAVIS[46], containing multiple real animal videos with mask annotations. For PlanetZoo, we choose 24 out of 249 total animals for testing and use the rest for validation and training. The testing dataset includes diverse articulated animals from unseen categories, including multiple quadrupeds, bipeds, birds categories,

---

[2]https://ambientcg.com/

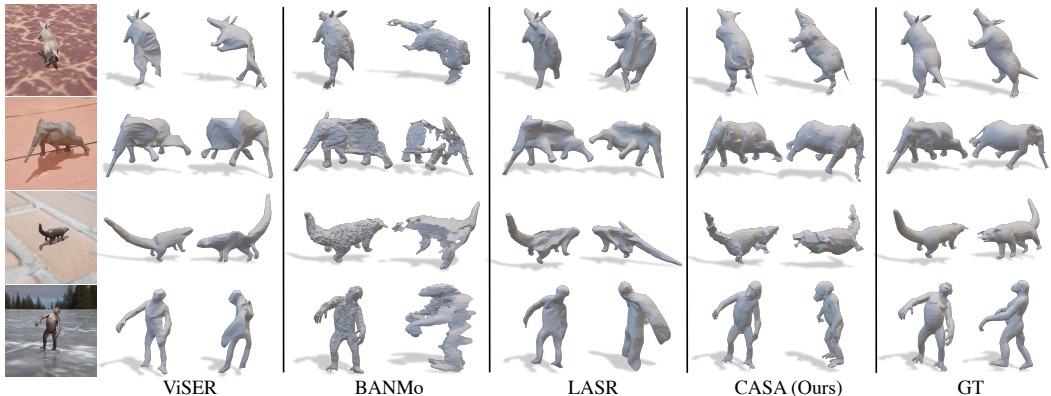

Figure 5: **Qualitative comparison of our method on PlanetZoo.**

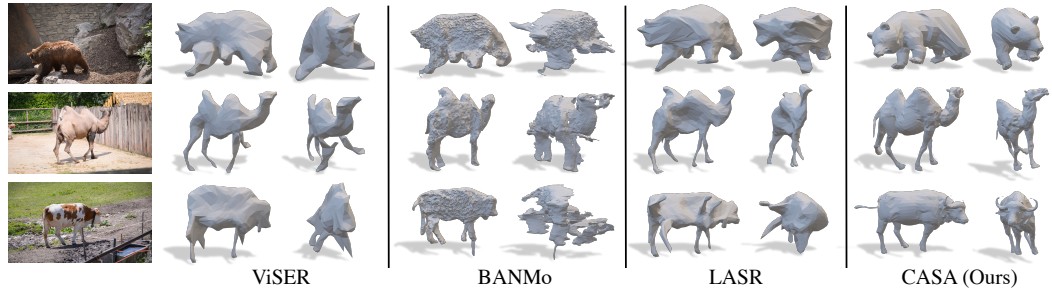

Figure 6: **Qualitative comparison of our method on real-world videos from DAVIS.**

as well as unseen articulated topology such as pinnipeds.

**Metrics.** We measure the reconstruction quality by Intersection over Union (IOU) and Chamfer distance, as well as skinning distance, joint distance and re-animation quality on PlanetZoo.

1) *mean Intersection Over Union(mIOU)* measures the volumetric similarity between two shapes. We voxelized the reference and the predicted shape into occupancy grids and calculate the IOU ratio.

2) *mean Chamfer Distance(mCham)* computes the bidirectional vertex-to-vertex distances between the reference and the predicted shape.

3) *Joint* is the symmetric Chamfer Distance between joints. We evaluate CD-J2J following [72]. Given a predicted shape, we compute the Euclidean distance between each joint and its nearest joint in the reference shape, then divide it by the total joint number.

4) *Skinning* distance measures the similarity between the skinning weights. We first extract vertices associated with each joint using skinning weight. For each pair of joints from GT and prediction, we calculate the Chamfer distance between their associated vertices. Finally, we exploit the Jonker-Volgenant algorithm to find the minimum distance matching between prediction and reference.

5) *Re-animation* measures how well we can re-pose an articulated shape to a target shape. Specifically, we minimize the Chamfer Distance between the reference shape and the predicted shape by optimizing the joint angles of each bone for skeletal shape, or the rigid transform for control point-based shape. We consider this a holistic metric that jointly reflects the quality of shapes, skinning, and skeleton.

**Baselines.** We compare with state-of-the-art approaches for monocular-video articulated shape reconstruction ,including LASR [74], ViSER [75], and BANMo [76][3]. Similar to our method, they utilize 2D supervision of videos for training including segmentation masks and optical flows. We download the open-source version of them from GitHub. For the input data, we use our ground truth silhouette either in PlanetZoo dataset or DAVIS[46] dataset. We follow the optimization scripts in their code and get the baseline results.

3D reconstruction from a monocular video inevitably brings scale ambiguities. To alleviate the issue of scale differences, for every predicted shape among all the competing baselines, we conduct a line

---

[3]We thank Gengshan Yang for sharing the code and providing guidance in running the baseline algorithms.

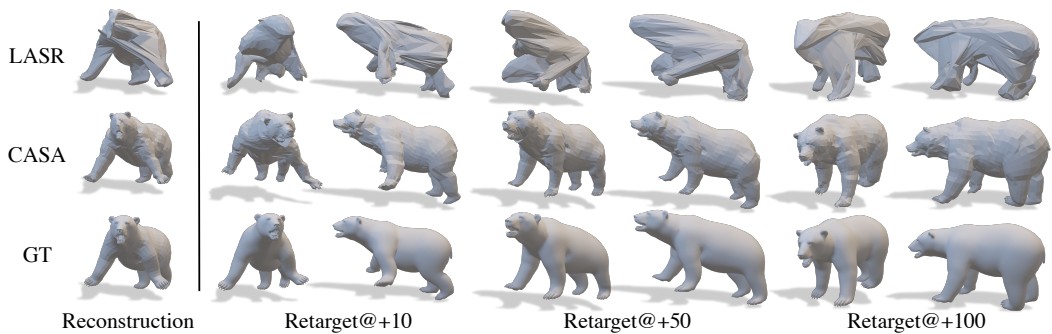

| LASR |
| CASA |
| GT |

| Reconstruction | Retarget@+10 | Retarget@+50 | Retarget@+100 |

Figure 7: **Reanimation results.**

search to find the optimal scale that maximizes the IOU between the reference and the predicted shape.

## 5.2 Results on PlanetZoo

We present quantitative results on PlanetZoo in Tab. 1. CASA achieves higher performance on metrics mIOU and mCham, demonstrating a shape with higher fidelity. Our approach also outperforms other competing methods in reanimation, demonstrating the superior performance of holistic articulated shape reasoning and the potential for downstream reanimation tasks. CASA compares less favorably to LASR in the skinning quality. We conjecture this is due to additional degrees of freedom benefits brought by joint control point articulation.

Fig. 4 and Fig. 5 depict a few qualitative examples of all competing algorithms on PlanetZoo. We show the reconstructed mesh from both the camera view and the opposite view. The figure shows that CASA can recover accurate shape across various view angles under partial observation. In contrast, baselines fail to reconstruct reliable 3D mesh when the objects are partially observed. In addition, CASA also produces meshes with higher fidelity and local details in the visible region.

## 5.3 Real-world reconstruction

We demonstrate qualitative results of the competing algorithms on the real-world animal video dataset DAVIS [46] in Fig 6 and Fig. 4. This figure shows that both our method and baselines can get good reconstruction results from the camera perspective. But both two baselines fail to reconstruct those unseen parts. In contrast, CASA can reliably reason the shape and articulation in the unseen regions, thanks to its symmetry constraints, skeletal parameterization, and 3D template retrieval.

## 5.4 Reanimation

We now show how the reconstructed articulated 3D shape can be retargeted to new poses. Given an inferred skeletal shape and the target GT shape at a different articulated pose, we apply an inverse kinematic to compute the articulated pose to reanimate the inferred shape. Specifically, we optimize the articulated transforms such that the reanimated mesh is as close to the target in Chamfer distance. Joint quaternions are optimized for skeletal mesh, and the articulation transforms are optimized for the control-point-based method [74]. Fig. 7 shows a comparison between the retargeted meshes of our methods and LASR on PlanetZoo, with a GT target mesh as a reference. We observe that our retargeted meshes look realistic and accurate and preserve geometric details. Despite having more degrees of freedom in reanimation, LASR fails to produce realistic retarget results. Tab. 1 reports a quantitative comparison in chamfer distance between the retargeted and the reference mesh, demonstrating our approach outperforms competing methods by a large margin.

Table 2: Ablation study on energy terms[4]

| Method | mIOU ↑ | mCham ↓ | Skinning ↓ | Joint ↓ |
|---|---|---|---|---|
| CASA | **0.512** | 0.053 | 3.288 | **0.089** |
| w/o $E_{mask}$ | 0.011 | * | * | * |
| w/o $E_{flow}$ | 0.427 | 0.350 | 3.290 | 0.636 |
| w/o $E_{symm}$ | 0.387 | * | 4.008 | * |
| w/o $E_{smooth}$ | 0.449 | **0.041** | **3.228** | 0.330 |

Table 3: Ablation study on optimization settings.

| Method | mIOU ↑ | mCham ↓ | Skinning ↓ | Joint ↓ |
|---|---|---|---|---|
| CASA | **0.512** | **0.053** | 3.288 | **0.089** |
| w/o offset | 0.144 | 0.343 | **3.174** | 0.783 |
| w/o disp. field | 0.235 | 0.055 | 3.218 | 0.262 |
| w/o skeleton | 0.372 | 0.060 | 3.245 | N/A |
| w/o scaling | 0.299 | 0.072 | 3.329 | 0.356 |

Table 4: Ablation study on our retrieval strategy.

| Method | mIOU ↑ | mCham ↓ | Skinning ↓ | Joint ↓ |
|---|---|---|---|---|
| CASA | **0.512** | **0.053** | **3.288** | **0.089** |
| Retrieval-only (CLIP) | 0.217 | 0.448 | 4.108 | 0.303 |
| Retrieval (ImageNet) | 0.111 | 0.972 | 4.295 | 0.813 |

Table 5: Ablation study on initialization settings.

| Method | mIOU ↑ | mCham ↓ | Skinning ↓ | Joint ↓ |
|---|---|---|---|---|
| CASA (retrieval init) | **0.512** | 0.053 | 3.288 | **0.089** |
| k-means rigging init | 0.305 | 0.452 | 3.336 | 0.659 |
| Fixing skinning weight | 0.433 | **0.052** | **3.235** | 0.318 |

## 5.5 Ablation study

We provide an ablation study to demonstrate the efficacy of each design choice in our framework.

**Energy terms.** In Tab. 2, we ablate different terms of our energy function. Specifically, we consider the mask consistency, flow consistency, symmetry and smoothness regularization separately. We find that the mask part is crucial for the performance, while the optical flow part also improves the framework results. Removing the symmetry offset term results in performance degradation since this term plays a vital role in regularizing the neural offset, which can alleviate issues brought by the ambiguity of single-view monocular video. The smooth term leads to better qualitative results.

**Optimization.** We compare different optimization settings in Tab. 3. The results show that: 1) using neural offset field is superior to per-vertex offset or not deforming shape (CASA vs. w/o offset vs. w/o disp. field); stretchable skeleton helps (CASA vs. w/o skeleton); removing shape scaling step leads to performance degradation (CASA vs. w/o scaling).

**Retrieval strategies.** In Tab. 4, we test different retrieval strategies. Our result demonstrates that CLIP is the preferred backbone for retrieval, most likely due to training with significantly richer semantic information than ImageNet pretrained models.

**Initialization.** We test different initialization strategies for skinning weights in Tab. 5. We replace the rigging from the retrieved animal by using k-means for initializing weights. The comparison in the table shows that high-quality rigging weight initialization is essential for good shape predictions. The results of sphere initialization also confirm the necessity of the proposed retrieval phase.

**Stretchable bone.** Fig. 8 shows qualitative results with/without stretchable bone parameterization. Compared to merely deforming each vertex, stretchable bone deformation ensures global consistency and smoothness. As noted in the figure, without stretchable bones, we see discontinuity at the nose of the animal and a mismatch between the lower and upper mouth.

## 5.6 Limitations

While demonstrating superior performance in animal reconstruction, our method has a few remaining drawbacks. Firstly, the source asset bank restricts the diversity of the retrieved articulation topology. Bone length optimization partially alleviates such limitations through "shrinking" bones. Still, it cannot add new bones to the kinematic tree,

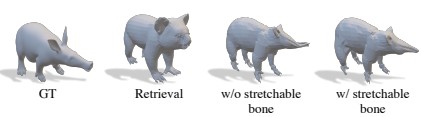

GT    Retrieval    w/o stretchable bone    w/ stretchable bone

Figure 8: Stretchable bone

e.g., we cannot create a spider from a quadrupedal template. Secondly, our method so far does not impose the constraint that bones are inside the mesh. We plan to tackle such challenges in the future.

## 6 Conclusion

In this paper, we propose CASA, a novel category-agnostic animation reconstruction algorithm. Our method can take a monocular RGB video and predict a 4D skeletal mesh, including the surface, skeleton structure, skinning weight, and the joint angle at each frame. Our experimental results show that the proposed method achieves state-of-the-art performances in two challenging datasets. Importantly, we demonstrate that we could retarget our reconstructed 3D skeletal character and generate new animated sequences.

**Acknowledgements.** The authors thank Vlas Zyrianov and Albert Zhai for their feedback on the writing. The project is partially funded by the Illinois Smart Transportation Initiative STII-21-07. We also thank Nvidia for the Academic Hardware Grant.

---

[4]∗: extremely large value

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
