# OpenReview forum: "CASA: Category-agnostic Skeletal Animal Reconstruction"
_NeurIPS.cc/2022/Conference — NeurIPS 2022 Accept_

### Official Review · Reviewer_4EwH · 2022-07-09

**Rating:** 7
**Confidence:** 5
**Soundness:** 3 good
**Presentation:** 3 good
**Contribution:** 3 good

**Summary:**

The authors propose CASA, a method for recovering 3D shape and skeletal movements from monocular videos. Given a video, it first retrieves a shape and a skeleton from a library with 200+ animals, then optimizes both shape and articulations with differentiable rendering. Results are shown synthetic real datasets, with applications in re-posing. A new dataset PlanetZoo with 200+ animated animals is introduced.

**Questions:**

- How many categories does the dataset contain? What are the categories?
- How are the root node transformations represented and initialized? Are they the same as LASR/ViSER?
- During model retrieval, how to sample viewpoints when rendering the 3D model? How many viewpoints are needed? How long does it take to sampling and matching? Also, pairwise matching should produce SxNxT scores, where S is the number of shapes, and N is the number of renderings per shape, and  T is the number of frames. Then how is T and N marginalized?


**Limitations:**

Yes.

**Strengths And Weaknesses:**

**Strengths**
- The proposed PlanetZoo dataset is interesting, as I'm not aware of a dataset of similar size (>200 animals) and quality (with texture, skeleton and deformation). It has potential to be used to evaluate and benchmark the performance of animal 3D reconstruction algorithms. From that perspective, it would be helpful to highlight the features of the dataset, and compare with the existing ones (such as deforming things 4D).
- The method is sensible. It leverages 3D shape and skeletal priors of specific categories to improve the reconstruction quality. Since the library is large, the method can be applied to a wide range of animal categories.
- The usage of skeleton structure also helps the re-posing application and user controllability.

**Weakness**
- Quality of results. The results are not faithful to the input data even test-time optimization is used. For instance in Fig.7, the reference cow image does not contain a horn but the reconstruction has horns, possibly from the template being retrieved. This is not desirable. I'm also confused as I did not notice a term that prevents the template shape to change when there is disagreement with the image evidence during optimization. This makes it unclear whether the optimization works properly.
- The experimental comparison can be made fairer. LASR/ViSER does not have access to a 3D shape library. Proper baselines would be methods that use a 3D shape template, such as ACSM [A], ACFM, or proving LASR a 3D template.
- There are some missing details (see questions). The one I'm most concerned about is the initialization of root transformations, which is crucial for reducing the ambiguity between shape and deformation. For instance, if the heading direction is rotated by 180 deg at initialization (the head becomes the tail), the optimization might focus on deforming the shape without correcting the heading.
- Ablation study is missing in the main paper. How well does retrieval perform? As it is a major contribution, a quantitative evaluation and thorough analysis is expected. For the ablation study on flexible bone model, a qualitative comparison is desired as numbers does not give much insight in this context.
- Fig. 3 is slightly misleading as it is not clear which parameters are per-frame and which are per-video. The subtitles also conflated per-video deformation (stretch and deform) vs per-frame deformation (rigging), vertex deformation (deform) vs bone deformation (stretch).

**Other related work**
- [A] Articulation-aware canonical surface mapping. CVPR 20.
- [B] Watch It Move: Unsupervised Discovery of 3D Joints for Re-Posing of Articulated Objects, CVPR 22.

---

> ### Author Response · Authors · 2022-08-02
> **Responses to Reviewer 4EwH**
>
> ### Horns in Figure 7 and optimization
>
> Thank you for pointing this out! We agree with the reviewer that the remained horn is not desirable.
>
> **Shape optimization**: There are two possible ways to deform the canonical shape in the optimization process: 1) the neural displacement field described in Line 208 - 219 of our paper. 2) the changes in bone length. Stretchable bone cannot handle this case as no "bones" exist in the horn component. However, the neural displacement field in our paper provides a fine-grained shape deformation, whose flexible parameterization can remove the horn by providing the correct image-based evidence.
>
> **Root causes**: However, we found that the mask and flow energy are small in practice for this case. This is because of 1) the tiny size of the horn region and 2) the majority of the horn regions are rendered inside the mask. Both prevent it from providing strong signals to guide the large deformation. We believe expanding our framework to include photometric loss (minimizing RGB appearance) or even feature-metric loss (minimizing feature difference) will help to overcome this issue. We will add this into the limitation discussion and leave it as a future direction.
>
> As shown in other qualitative results (e.g. Figure 6, 7 and supp Figure 2), we want to highlight that most of our recovered shapes have faithful and realistic shapes and poses after optimization, suggesting the efficacy of optimization.
>
> ----------------------------------------------------------------------------------
>
> ### Comparison against ACFM
>
> We compared our method against the template-based ACFM. Note that ACFM is category-specific; hence we only compare all the quadruped animals in the PlanetZoo testset to ensure a fair comparison. We report the results in supp Table 5. Results show that CASA significantly outperforms ACFM even though the later has a network component trained specifically for quadruped animals (mIOU: CASA 0.499 vs ACFM 0.234).
>
> ----------------------------------------------------------------------------------
>
> ### Our retrieval + other baselines
>
> Due to time constraints, we have not yet completed adjusting the LASR code to use our retrieval template. Our final version will compare CASA-retrieval + LASR vs. the entire CASA pipeline. This comparison will mainly demonstrate the efficacy of our optimization pipeline.
>
> That said, we also think the current comparison against template-based and template-free methods is fair, as CASA's 2D-3D retrieval is a crucial part of our contribution. Yet, being template-free is one core claim in many baselines; Augmenting other baselines with our proposed retrieval results in a different approach for comparison.
>
> ----------------------------------------------------------------------------------
>
> ### Root initialization
>
> For real-world data or synthetic settings without GT camera poses, we initialize the root transform for each frame by minimizing the rendering mask loss at a coarse level while treating the rest as rigid. A diverse set of random initial root rigid transforms are used for repeated optimization and the root transformation at the lowest mask loss is selected. There are indeed local optimal as described by the reviewer (180 flips), but our multi-init procedure helps get rid of most cases, as correct alignment still offers lower loss. For synthetic data with given camera poses, we directly use them as root transform initialization (the same is applied for all competing baselines for a fair comparison).
>
> After the initialization, the root transformation is jointly optimized with other parameters (bone joint angle, displacement field, etc.) by minimizing the proposed energy function.

---

> > ### Author Response · Authors · 2022-08-02
> > **(Cont.) Responses to Reviewer 4EwH**
> >
> > ### Ablation study
> >
> > * **Retrieval strategies**: We show the full retrieved results in supp Table 7 – including Top-1 for each animal. It’s hard to quantitatively evaluate how good retrieval performance is, as our testing set consists of novel categories. That said, we compare the final reconstruction quality between the proposed retrieved skeletal shape vs. other init strategies in supp Table 4. In particular, we include: 1) initializing skinning weight by k-means (mIOU: retrieval init 0.435, k-means init 0.305), 2) initializing shape by sphere (mIOU: retrieval init 0.435, sphere init 0.277). These results demonstrate the necessity of retrieval, as initializing by retrieved skeletal shapes boost the optimization performance by a large margin. In addition, we add the quantitative results on retrieved shapes without optimization in supp Table 3. The experiments show that retrieval does provide reasonable results, since the retrieved shapes achieve relatively good IoU and chamfer distance values without optimization.
> > * **Stretchable bones**: We also add the qualitative comparison with/without the flexible bone parameterization in supp Figure 4. Due to the time limits, we did not complete a quantitative evaluation. We will include this in our camera ready.
> >
> > ------------------------------------------------------
> >
> > ### Sample viewpoints
> >
> > Our 3D asset consists 225 animal categories. We render 180 realistic frames of each animal under different poses from different viewpoints.
> >
> > We marginalize the similarity of a query video as follows: 1) given a frame of the video, find the closes image over each animal category using CLIP and store its image similarity score. 2) calculate the similarity score between the video and a given animal category by taking the sum of the similarity between each frame and that animal; 3) take the category with the highest similarity score. To summarize, our retrieval procdure calculate the following function:
> >
> > $$\arg\max_j \sum_t \max_v  \langle g_\mathrm{CLIP}(\mathbf{I}_t), g_\mathrm{CLIP}(\pi(\mathbf{s}_j, \mathbf{q}_v)) ) \rangle,$$
> >
> > where ${\mathbf{I}_{1...T}}$ is the input video, $\mathbf{s}_j$ is the $j$-th animal shape, $g_\mathrm{CLIP}$ is the image embedding network of the CLIP model and $\pi(\mathbf{s}_j, \mathbf{q}_v)$ is the photo-realistic rendering of the articulated shape $\mathbf{s}_j$ at a randomized skeletal pose $\mathbf{q}_v$.
> >
> > ------------------------------------------------------
> >
> > ### Dataset statistics
> >
> > We provide a detailed comparison between PlanetZoo and other popular dynamic 3D dataset, including DeformingThings4D and SAIL-VOS 3D, in the table below.
> >
> > Dataset | Category | Character | Frame | Realistic texture | RGB | Depth | GT camera | GT mask | GT mesh |
> > :-----| :----: | :----: | :----: | :----: | :----: | :----: | :----: | :----: | :----: |
> > DeformingThings4D | 31 | 147 | 122,365 | No | No | No | No | No | Yes |
> > SAIL-VOS 3D | 10 | multiple | 111,654 | No | Yes | Yes | Yes | Yes | Yes |
> > PlanetZoo | 249 | 249 | 44,820 | Yes | Yes | Yes | Yes | Yes | Yes |

---

> > > ### Comment · Reviewer_4EwH · 2022-08-08
> > > **Thanks for providing a rebuttal.**
> > >
> > > Most of my questions are answered adequately. I'd like to raise score to 7. The interesting part of the paper is the 3D skeletal model retrieval given a large database, which provides reasonable constraints when the target object falls roughly within the database. The remaining concern is that the method does not appear faithful to the data (horns of the cow in Fig. 4), which could be  due to either lack of observation or unnecessary regularization terms, which needs further discussion.
> > >
> > > Suggestion on figures
> > > - Please visualize the retrieved model together with the reconstruction (similar to lasr did), so that readers understand how much the shape is updated by optimization.
> > > - Lasr results in Fig. 8 is inconsistent with Fig.7
> > > - Fig 6. Note casa/lasr apply symmetry constraint on the occluded body part, but banmo/viser baseline do not. This should be better explained.

---

### Official Review · Reviewer_ziQq · 2022-07-11

**Rating:** 7
**Confidence:** 4
**Soundness:** 2 fair
**Presentation:** 3 good
**Contribution:** 3 good

**Summary:**

This paper presents a category-agnostic character animation reconstrucion from a casual video input. To alliviate the ill-posed nature of the problem, this work first queries the closest template model from the database using CLIP features. As the following inverse graphics optimization stage can warm start with the retrieved template, the proposed approach produces better results both qualitatively and quantitatively over prior methods. Additionally, the paper introduces a large amount of synthetic dataset for qualitative evlauation of predicted attributes with diverse categories.

**Questions:**

- Please answer to the comments above.
- It is not clear if CLIP is necessary for this retrieval task as text is not involved at all. It would be great if the use of various feature backbone including CLIP and ResNet pretrained with ImageNet is evaluated.
- From the exposition, it is not clear how to compute CLIP feature from videos (CLIP only provides embedding per frame). Please elaborate.

Other comments:
- L43: The neural inverse graphics framework in general has been extensively used in prior works, and not novel. Please state more concretely what is novel over the prior works in terms of the optimization.
- L231: Please add citation to Cobra-tools.

**Limitations:**

The limitation is discussed, but its societal impact is not.

**Strengths And Weaknesses:**

This paper has the following strengths:
- This work presents an interesting use of CLIP feature. To retrieve the closest animal template from the database, the CLIP features are extracted from both input video and synthetically rendered database. This type of semantic-based retrieval is more general and flexible than hand-crafted descriptors for the retreival task.
- The paper presents an impressive qualitative results even from videos in the wild.
- The proposed dataset would be a great resource for community for both training and evaluation to assess the accuracy of predicted attributes from diverse categories.

There are several weaknesses of this work:
- The qualitative results of baseline methods (LASR, ViSER) are substantially worse than what’s presented in their original papers. I’m wondering if the results are cherrypicked or their code was not properly run. I would highly recommend reaching out to the authors of these papers to confirm that these results are expected. Please answer to this in the rebuttal to prove that the experiments are credible.
- L58-60: I’m very confused about the notion of generalization here. The paper also discuss train/test split in L258. However, as far as I understand, this work presents an instance-specific training and there is no cross-instance generalization. Please clarify.

Overall the paper presents an interesting approach and the results are impressive. However, the aforementioned concerns prevent me from giving a higher score at this stage.

---

> ### Author Response · Authors · 2022-08-02
> **Responses to Reviewer ziQq**
>
> ### Qualitative Comparison
>
> We contacted the author of LASR/ViSER regarding reproducing their results.
> * **LASR**: We verified that we fully reproduced LASR results reported in their table through the email discussions. Our comparison is also conducted fairly.
> * **ViSER**: Our reported qualitative and quantitive results in the submission are worse than ViSER results for two primary reasons: 1) the master Github repo in ViSER had a bug by NeurIPS 2022 deadline, which was fixed in June; We updated ViSER results with the latest master repo. 2) ViSER reported the qualitative shapes used a larger smooth hyper-parameter (0.25) for better visual quality. This setting differs from the config used in the paper's quantitative evaluation. For consistency and fair comparison, we reported the qualitative and quantitative results using the same hyper-parameters. The authors have verified our reported qualitative results in the revised submission on BADJA dataset.
>
> --------------------------------------------------
>
> ### Clarification of train/val split.
>
> The inverse graphics stage is training-free test-time-optimization. Hence as the reviewer points out, there is no concern regarding cross-instance generalization. However, our retrieval stage currently relies on retrieval from an existing asset bank as a template. To demonstrate category-agnostic reconstruction ability, it is crucial to ensure the assets do not overlap with testing animals at both instance and category levels. In other words, testing samples should come from unseen categories/instances or even include unseen topologies. In addition, the optimization stage also has several hyper-parameters. Our dataset split also allows us to tune hyper-parameters in the training set. The testing dataset is only used for evaluation purposes. Hence, train/test split is necessary for our dataset/benchmark.
>
> --------------------------------------------------
>
> ### Necessity of CLIP
>
> CLIP has been trained with significantly richer semantic information than ImageNet pre-trained models. Such information is encoded in a rich text corpus and allows us to capture complicated relationships between images from animals. In practice, we found it is crucial for retrieval performance. Specifically, compared against models pre-trained on ImageNet, we found CLIP retrievals provide a better skeletal shape (see **supp Table 3** for a comparison). The results show that CLIP is the preferred retrieval backbone than ImageNet pre-trained models (mIOU: CLIP 0.217, ImageNet pre-trained 0.111).The retrieved animal also agrees with humans' common sense.
>
> --------------------------------------------------
>
> ### CLIP features computation
>
> We provided details of CLIP feature computation and retrieval in **supp Sec.H**.
>
> --------------------------------------------------
>
> ### Inverse graphics
>
> To our limited knowledge, we are the first work to incorporate skeletal and stretchable bone parameterization in generic articulated shape reconstruction. The topology of the skeleton tree is difficult to directly recover using inverse graphics, especially when the shape is jointly optimized. We innovatively use template retrieval and bone-length optimization to overcome this challenge, making it possible to optimize shape and skeleton jointly.

---

> > ### Comment · Reviewer_ziQq · 2022-08-08
> > **Re: Responses to Reviewer ziQq**
> >
> > Thank you for the detailed responses as well as additional experiments to address the raised concerns.
> >
> > Now that my major concerns are cleared, I raised my score to accept. However, as R1 mentioned, the loss functions on inverse graphics are not substantially novel and I would strongly recommend toning down the contribution claim on neural inverse graphics and better clarifying the novelty. Also since the proposed approach does per-instance optimization, the generalization claim in the sentence of L58-60 does not make sense. I would highly recommend removing it. Thanks!

---

### Official Review · Reviewer_pF6x · 2022-07-12

**Rating:** 4
**Confidence:** 3
**Soundness:** 2 fair
**Presentation:** 2 fair
**Contribution:** 2 fair

**Summary:**

This paper focuses on the problem of 3D reconstruction of animals from a monocular video. The proposed method is claimed to be Category-agnostic, which means they can deal with different categories such as dogs, horses, but they are all quadruped. The major contributions is the category-agnostic reconstruction which is realized by first retrive a 3D template model from an asset. Then the template model is deformed and optimized to fit to the input video.

**Questions:**

I have some questions or confuse on the some technical details of the proposed method,
1) How to optimize both the bone length, joints angles, skinning weights together? Do we need to optimize one while fixing others? Furthermore, what is the improvement of optimizing skinning weights, what if the skinning weights are not optimized, but using those from retrived template?

2) How is the optimization initialzed? How can we achieve the initial fitting to images or videos, including initial global pose?

The authors might also want to include the response to the issues I raised in Weakness above.

**Limitations:**

The major limitation is lack of technical contribution. Using the pre-trained CLIP model as features to retrive the 3D template is good, but I'm not sure this could be claimed as technical novelty. And the following optimization is also pretty standard. The authors should point out what are the major technical contributions that really stands out.

**Strengths And Weaknesses:**

Strengths:
First, having a good initial template to start with the following up optimization will certainly reduce the deformation space and make the optimization to be more feasible. They have demonstrated better visualization results by adopting this retrived template model. The idea is pretty easy to understand and the paper is organized and written well.

Weakness:
1) Obtaining this template model by retrival using the pre-trained CLIP model is more or less an engineering work. I don't think this could be claimed as a technical contribution. The performance is improved mainly due to this retrived template, while the compared approaches start with some general shape for example, a sphere.
2) The optimization pipeline or the loss function is not novel. It is pretty much a standard optimization, I'm a bit confused what the authors want to claim on this optimization problem. In addition, the skeletal representation of articulated models, they are just standard way of dealing with those kinds of animals. I'm not sure what the authors want to emphasize on this.
3) Missing important comparison:  -- BANMo: Building Animatable 3D Neural Models from Many Casual Videos.

---

> ### Author Response · Authors · 2022-08-02
> **Responses to Reviewer pF6x**
>
> ### "All quadruped animals in paper"
>
> We respectfully disagree that “they are all quadruped”: please see supp video (02:07) and supp Figure 1 for ostrich; supp video (02:10) for chimpanzee; supp Figure 2 and supp video (01:57) for seal.
>
> --------------------------------------------------
>
> ### Novelty of template retrieval
>
> We respectfully disagree that our retrieval is “*more-or-less an engineering work*.” We will discuss the contributions of our proposed retrieval based on technical novelty and the impacts on 4D reconstruction.
>
> * **Novelty**: 2D-3D articulated retrieval is underexplored. Our CLIP-based approach to such a problem, to our limited knowledge, has not been explored before.
> * **Impact**: Prevailing template-based methods are limited to one of a few templates, often in a category-specific manner [a, b]. This restricts the method from achieving comparable results in the category-agnostic setting. Our approach closes this gap by enabling us to initialize from various fine-grained templates for articulated reconstruction. As suggested by Reviewer ziQq, and Reviewer 4EwH, it allows the reconstruction method to leverage 3D priors and offer stronger generalization ability and improved reconstruction quality.
>
> [a] Kulkarni, Nilesh, et al. "Articulation-aware canonical surface mapping." CVPR 2020
>
> [b] Kokkinos, Filippos, and Iasonas Kokkinos. "Learning monocular 3D reconstruction of articulated categories from motion." CVPR 2021
>
> --------------------------------------------------
>
> ### Contributions of Optimization
>
> We want to stress the key difference between our optimization vs. previous works lies in the parametric models.
> * We adopt the skeletal shape model. It is critical since this model induces bone constraints for nonrigid motion and allows us to conduct realistic re-animation using the reconstructed shape. As a comparison, most previous category-agnostic 4D reconstruction (e.g., LASR, VISER, BANMo) uses a mixture of rigid transforms as their nonrigid kinematic model. During inference, they directly optimize rigid body transformation without considering the bone constraints, which could result in less appealing nonrigid motion.
> * We also leverage a stretchable bones parameterization and a neural-parametric vertex deformation model, offering more realistic and smooth shape deformation.
>
> As noted in the paper, although the two techniques are used in graphics for animation simulation and modeling, applying them to category-agnostic articulated objects is highly non-trivial and innovative to our community. Our superior results also justify the importance of such technical choice.
>
> --------------------------------------------------
>
> ### Optimization details
>
> * **Optimizer**: We optimize all the parameters jointly using Adam. No alternative optimization is used, thanks to a good initialization from the retrieval stage.
> * **Skinning weight**: Compared to fixing skinning weight, updating skinning weights would allow more flexibility when the initial skinning weight is of low quality. Since our retrieval strategy provides high quality skinning weight initialization in most cases, the metrics would not show significant differences (mIOU: optimizing skinning 0.435, fix skinning 0.433).
> * **Initialization**: Our shape/rigging/bone parameters are initialized with all the corresponding parameters from the template. Joint angles are initialized from a T-pose. Global camera poses (at object-centric-coordinate) are encoded as the root node transformation. The global pose is initialized by minimizing the mask loss at T-pose.
>
> --------------------------------------------------
>
> ### Contributions Clarification
>
> The key contributions of this paper are 1. we present **a diverse skeletal shape asset** (our dataset); 2. we revive template-based reconstruction using **a simple, effective and generalizable 2D-3D retrieval algorithm** based on a pretrained CLIP model; and 3. a novel **skeletal shape optimization** procedure. We show that through the three key components, we could push the articulated shape reconstruction quality to another level. We also introduced a new realistic simulation-based benchmark in the hope of bringing more vibrance to the community.

---

### Author Response · Authors · 2022-08-02
**General responses**

We thank the reviewers for their feedback and helpful suggestions. The reviewers agree our template retrieval is "*generalizable and flexible*" (Reviewer ziQq, 4EwH), "*impressive qualitative results*" (Reviewer pF6x, ziQq), and "*the great benefit to the community*" of our proposed dataset (Reviewer ziQq, 4EwH).

This comment summarizes the major revisions we make to our submission. We also reply to each reviewer's questions individually. We strongly recommend the reviewers and the ACs read both rebuttal comments and the revised submission. Please do not hesitate to ask follow-up questions during the reviewer-author discussion period.

-------------------------------------------------------

### Comparison studies.
We updated and added several baseline methods for comparison (VISER, BANMo, ACFM). **Table 1** and **Figure 6 and 7** summarized the results:
* We updated **VISER** results following the author's recent bug fixes in June 2022.
* We added **BANMo** as an additional template-free baseline.
* We reported (**A-CFM**) and compared it with CASA in supp Table 5 on quadruped animals testset.
* We contacted the author of **LASR, VISER, and BANMo** and validated that the comparison was fair and correct.

-------------------------------------------------------

### Technical details.
* We added details on the initialization strategy of CASA (**supp Sec.I**).
* We provided more details on the CLIP-based retrieval procedure in (**supp Sec.H**).
* We provided more information about the new dataset in (**supp Table 6**).

-------------------------------------------------------

### Ablation study.
We provided a more thorough ablation comparison and discussion, including quantitative performance against retrieval-based methods, fixing initial skinning weight. Please see **supp Table 3 and 4**.

-------------------------------------------------------

### Missing related works.
We have added all related works suggested by the reviewers in our rebuttal revision.

---

### Meta-Review · Area_Chair_Hiou · 2022-08-27

**Recommendation:** Accept
**Confidence:** Certain

**Metareview:**

The paper shows how to combine 3d model retrieval with an inverse graphics framework to recover 3D models of a diverse range of animals from video.   The paper also introduces a new dataset of 3D animals that is projected to be of value in future works.

While one reviewer considers the technical problem to be "an engineering work", the other reviewers, and the AC, consider that the implementation and experimentation of the effects of this novel (in this context) idea is valuable.

Based on calibration across other papers and reviews in this AC's stack, the average review score is generally inconsistent with the review text, even given the effective rebuttal.  I mention this only because a poster acceptance might seem at odds with average score, but of course the point of meta reviewing is to make a judgement which looks at more than average score.  The key decision that might be affected in this case is oral vs poster, so it is perhaps useful to clarify: an oral presentation needs to be of value to the broad NeurIPS community.  3D computer vision is an important subfield, and animal reconstruction is an emerging topic in the subfield, but the learnings of this paper remain essentially within a subfield, so I am confident that poster is the appropriate disposition of this paper.


**Award:**

No

---

### Decision · Program_Chairs · 2022-09-14

Accept